# Game Analysis of the Evolution of Local Government’s River Chief System Implementation Strategy

**DOI:** 10.3390/ijerph19041961

**Published:** 2022-02-10

**Authors:** Juan Wang, Xin Wan, Ruide Tu

**Affiliations:** 1School of Management, Wuhan Institute of Technology, Wuhan 430205, China; wangjuan@wit.edu.cn; 2Institute of Income Distribution and Public Finance, School of Taxation and Public Finance, Zhongnan University of Economics and Law, Wuhan 430073, China; gracewanxin@zuel.edu.cn; 3School of Information Management, Central China Normal University, Wuhan 430079, China

**Keywords:** River Chief System, evolutionary game, local government, evolutionary stabilization strategy, sewage discharge, Pareto optimal, Q53

## Abstract

As the executor of the River Chief System (RCS), local governments’ choice of implementation strategies directly affects the quality of regional water environment. The implementation of the RCS involves many interest subjects, and has gradually formed a game between enterprises’ sewage management and local governments’ RCS implementation strategies, and a game between the RCS implementation strategies of different local governments. The game behavior between the interested parties is long-term and dynamic in nature. Strategies such as reducing the cost of local governments’ implementation of the RCS and increasing the rate of sewage charges will lead to the evolution of the strategy set between enterprises’ treatment of sewage and local governments’ RCS implementation in the direction of {complete treatment of sewage, strictly enforcing the RCS}. Analysis of the evolutionary game model between the local governments reveals that strategies such as reducing the weight of economic indicators in local governments’ assessment, and increasing the material and spiritual rewards for implementing the RCS, will lead to the evolutionary game outcome of implementing the RCS between the local governments in the direction of {strictly enforcing the RCS, strictly enforcing the RCS}. The external effects of sewage discharge do not affect the evolution of the game system between the local governments.

## 1. Introduction

In order to alleviate a series of economic and social problems caused by water pollution, China fully implemented the River Chief System at the end of 2018 to strengthen water pollution prevention and control, comprehensively remediate black smelly water bodies, and improve the ecological and environmental governance system. Water environment is a typical public good, and water environment control depends not only on central policies but also on local governments’ considerations based on political and economic interests [1]. The RCS consists of the party and government leaders at all levels in China serving as River Chief, which are responsible for organizing and leading the management and protection of the corresponding rivers, lakes, and other waters.

The implementation of the RCS involves many interest subjects, and gradually leads to the formation of a game between enterprises’ treatment of sewage and local government’s RCS implementation strategy, and a game between the RCS implementation strategies of the local governments. It has been found that the environmental control strategies of local governments are associated with the cost of environmental regulation enforcement, the strength of penalties for inaction, the weight of environmental indicators in the performance appraisal system, and the expected pollution emission reduction [2,3]. In the literature on the analysis of the game between local governments and polluting firms, some studies have analyzed their regulation and rent-seeking, respectively, from the perspective of the Nash equilibrium under mixed strategies [4]. Others have analyzed the equilibrium outcomes of the firm–government game under different institutional arrangements [5], and some studies have analyzed how environmental regulation affects the direction of industry output equilibrium in the context of Cournot competition and oligopoly [6]. In the literature on environmental strategy games among local governments, some studies have analyzed rational government and inefficient environmental regulation from the perspective of the Pigou tax [7], whereas others have used evolutionary games to analyze the evolutionary paths and stabilization strategies of local governments in air pollution control [8].

The existing literature has rarely analyzed the impact of strategies between enterprises and local governments, and between local governments from the perspective of RCS. As a local government policy innovation, the RCS has its own unique factors. For example, most of the performance assessments of River Chiefs in each region are linked to the effectiveness of the implementation of the RCS, thus making it more politically risky in terms of advancing their position to focus only on economic growth and ignore water pollution. Therefore, local officials are more motivated to supervise sewage-discharging enterprises in their jurisdictions based on political promotion [9]. In addition to the effect of linking the implementation of the RCS to the evaluation of officials’ promotions, most regions also impose economic or political punishments, such as demotion and legal sanctions, for officials with poor implementation of the River Chief System. As a result, the policy system is ensured by the RCS. The characteristics of the institutional setting and task arrangement determine that the RCS is more authoritative and binding than previous water pollution management policies in China, and the policy is more sustainable [10,11]. The impact of the RCS on the game system and the considerations in designing the model are also different from those examined in previous studies. Therefore, in order to effectively model these dynamic and long-term game relationships, the current article uses an evolutionary game model to explore the evolutionary process of RCS implementation decisions between enterprises and local governments, and between local governments. The exploration of these strategic behaviors is necessary in the process of RCS reform, and helps to reveal the nature of water pollution management in China and improve the efficiency of the implementation of the RCS.

## 2. Evolutionary Model of Local Government and Firms

### 2.1. Evolutionary Model Assumptions and Notation

(1)Model assumptions

The central government does not uniformly specify the form of the RCS implementation, and each provincial government sets up different governance models according to its own conditions. The vast majority of provinces and municipalities take the results of the RCS work assessment as an important element in the comprehensive evaluation of officials. Local governments are responsible for the quality of water bodies within their jurisdictions, while accepting supervision from higher levels of government and the public. Enterprises are guided by relevant national regulations to discharge effluent in accordance with effluent standards, and local governments will levy effluent charges for their effluent discharges, with the aim of inducing water discharge enterprises to internalize external costs.

In order to reasonably set up the evolutionary game model between local governments and water discharge enterprises, drawing on existing studies [12,13], the following assumptions are made in this study. (I) The RCS assessment system is the same for areas under the jurisdiction of the same provincial government. (II) Under the strict arrangement of RCS, local governments are required to treat and improve existing waters and other various tasks in addition to supervising water discharge enterprises in their jurisdictions. Local governments must implement the RCS work arrangement, or they will face huge political and economic penalties. Similarly, water discharge enterprises in the jurisdiction must treat their own sewage. Of course, if the economic benefits obtained by the government from not implementing RCS are greater than that from implementing reforms, and government officials do not consider penalties such as promotion or demotion, they may also choose not to implement RCS. (III) The goal of water discharge enterprises based on the economic goal of cost minimization and complete treatment of sewage is to meet the water pollution discharge standards set by the government. (IV) All parameters are in the same policy cycle during the study sample period and will not change over time. (V) Due to the strict implementation of the RCS, the water pollution in the jurisdiction is controlled and the water quality significantly improved. (VI) It is assumed that the regulatory costs incurred by local governments when they do not fully implement the RCS (including enforcement costs and costs of treating effluent) are greater than the costs associated with completing strict implementation of the RCS (at this point, the costs incurred by local governments for deregulation, resulting in more effluent treatment, are greater than the enforcement costs reduced by less strict regulation).

(2)Description of symbols

Based on the description of the evolutionary game between the local governments and water discharge enterprises, the relevant parameters are set as follows:

$c_{A}$ represents the implementation costs incurred by the local government A for strict implementation of the RCS, including the operating costs and water pollution control costs. The operation cost includes the human, material and financial resources invested by the local government. It also includes the renovation of sewage discharge pipes and construction of facilities.

c is the cost of fully treating wastewater for companies in the jurisdiction of local government A.

$h_{1}$ is the amount of sewage discharged by enterprises in the jurisdiction of local government A after complete treatment of the sewage. Generally speaking, the enterprise has completely treated its own sewage and met the discharge standard, and the discharged sewage will also contain the present pollutants.

$h_{2}$ is the amount of sewage discharged by enterprises in the jurisdiction of local government A after incomplete treatment of sewage, where $h_{2}>h_{1}$.

θ is the rate of sewage discharge charges in the jurisdiction.

α is the weight coefficient of the RCS work assessment index in the performance appraisal of local government A, where 0<α<1.

β is the weight coefficient of the economic assessment index in the performance appraisal of local government A, where 0<β<1.

λ is a coefficient of the implementation of RCS by local government A. According to the previous discussion, 0<λ≤1.

δ is the coefficient of the strength of the water discharge enterprises to treat sewage in the jurisdiction, where 0<δ≤1.

η is a factor for the increased costs associated with the incomplete implementation of RCS by local government A, resulting in more effluent discharges, where η>1. The lack of full implementation of RCS by local governments can lead to more effluent discharges, and thus a corresponding increase in their pollution control costs for performance and other goals.

x is the percentage of water dischargers within the jurisdiction of local government A that choose to fully treat their effluent, and y is the proportion of River Chiefs at all levels in the jurisdiction of local government A who choose to fully implement RCS.

### 2.2. Model Analysis

In the context of RCS, the local government’s control of enterprise effluent discharge is divided into strict implementation of RCS work; high frequency regulation of enterprise discharge behavior and strict control of effluent discharge; and incomplete implementation of RCS work arrangements, low frequency regulation of enterprise discharge behavior, and relaxation of effluent discharge control. The strategy set of the local government for RCS implementation is {strictly enforcing the RCS, not strictly enforcing the RCS}. Local government finances are derived from the production activities of various enterprises, which is a very important part of local government performance. Under the Chinese decentralized system, political centralization and economic decentralization give local governments a great deal of adjudication power in economic matters, and the central government guides and incentivizes local government behavior through the performance appraisal system. Economic targets imply that local governments have to achieve their goals through various types of enterprise development, and water quality assessment targets under the RCS require local governments to reasonably regulate the amount of effluent discharged by enterprises. Therefore, the economic return of the sewage water enterprises and the volume of sewage discharged are important factors influencing the level of payment by local governments. The sewage water enterprises then form the following strategy set: {complete treatment of sewage, incomplete treatment of sewage}.

Under the RCS arrangement, the game behavior between a local government and a water discharge enterprise is a stochastic matching, and the mutual influence of the repeated game process, and the decision adjustment process between the two can be simulated by replicating the dynamic mechanism. When local government A chooses to strictly implement the RCS, a certain percentage of polluting enterprises will choose to fully treat their sewage, and, in turn, the amount of sewage discharged within local government A’s jurisdiction will decrease and water quality will improve. If the local government A chooses not to strictly implement RCS, a certain percentage of enterprises will choose not to fully treat water pollution, and, in turn, the amount of sewage discharged within the jurisdiction of local government A will increase, and water quality will deteriorate. If the local government A chooses to strictly implement RCS and the polluting enterprises also choose to completely treat sewage, polluting enterprises will incur the cost of completely treating sewage with the cost of reasonable sewage $c+\theta h_{1}$. The local government A will incur RCS implementation costs, i.e., the economic performance loss $\beta \left(c+\theta h_{1}\right)$, which is the economic loss of water discharge enterprises to completely treat sewage, the loss of environmental indicator assessment $\alpha h_{1}$ discharged by polluting enterprises, and the gain of allowable sewage charges $\theta h_{1}$.

When local government A chooses to strictly enforce the RCS and polluting enterprises choose not to fully treat the sewage, the polluting enterprises will incur part of the cost of treating the sewage (at this point the cost of treating the sewage is mainly related to the strength of the enterprise to treat the sewage) and the cost of discharging sewage $\delta c+\theta h_{2}$. The local government A will incur RCS implementation costs, the economic performance loss $\beta \left(\delta c+\theta h_{2}\right)$, the loss of environmental indicators assessment $\alpha h_{2}$, and the gain of allowable sewage charges $\theta h_{2}$.

When local government A chooses not to strictly implement RCS and polluting enterprises choose to completely treat wastewater, polluting enterprises will incur the cost of completely treating wastewater and the cost of reasonably discharging water. At this time, the enterprises’ cost of discharging water is reduced due to less strict regulation, the reduction is related to the intensity of regulation, and the total cost is $c+\theta \lambda h_{1}$. In addition, the local government A cost will be increased and the total cost is set to $\eta c_{A}$, the economic performance loss $\beta \left(c+\theta \lambda h_{1}\right)$, environmental indicators assessment loss $\alpha h_{1}$, and the gain of allowable sewage charges $\theta \lambda h_{1}$. The gains at this point are partially lost due to local government deregulation, and the strength of the loss is directly related to the strength of RCS enforcement.

When local government A chooses not to strictly enforce the RCS and polluters choose not to fully treat the effluent, polluters will incur a portion of the cost of treating the effluent and the cost of discharging the water, $\delta c+\theta \lambda h_{2}$. Local government A will incur RCS implementation costs, the economic performance loss, environmental indicator assessment losses $\alpha h_{2}$, and the gain of allowable sewage charges $\theta \lambda h_{2}$. The repetition is played by randomly selecting participants in two groups of local governments A and water discharge enterprises in their jurisdictions. In the 2 × 2 asymmetric repeated game, the payment matrix of its stage game is shown in Table 1.

The proportion of polluters choosing to fully treat the sewage is set to x and the proportion of River Chiefs choosing to strictly enforce RCS is set to y. This study uses a replicated dynamic model to simulate a finite rational repeated game process between polluters and local government A.

The expected benefits for polluters choosing to treat their wastewater completely versus incompletely are as follows:(1)$$U_{1}=y(-c-\theta h_{1})+\left(1-y\right)\left(-c-\theta \lambda h_{1}\right)$$
(2)$$U_{2}=y(-\delta c-\theta h_{2})+\left(1-y\right)\left(-\delta c-\theta \lambda h_{2}\right)$$

The average expected benefits of wastewater treatment for polluters are as follows:(3)$${\bar{U}}_{12}=xU_{1}+\left(1-x\right)U_{2}$$

The replicated dynamic equation for the polluting firm choosing to completely treat the effluent is as follows:(4)$$F\left(x\right)=\frac{dx}{dt}=x(U_{1}-{\bar{U}}_{12})=x\left(1-x\right)\left(U_{1}-U_{2}\right)$$

Bringing Equations (1) and (2) into Equation (4), the following can be obtained after calculation:(5)$$F\left(x\right)=x\left(1-x\right)\left[\theta \left(h_{2}-h_{1}\right)\left(y-y\lambda +\lambda \right)-c\left(1-\delta \right)\right]$$

The expected benefits for River Chiefs choosing to strictly implement RCS and not strictly implement RCS are as follows:(6)$$U_{A1}=x\left[-c_{A}-\beta \left(c+\theta h_{1}\right)-\alpha h_{1}+\theta h_{1}\right]+\left(1-x\right)\left[-c_{A}-\beta \left(\delta c+\theta h_{2}\right)-\alpha h_{2}+\theta h_{2}\right]$$
(7)$$U_{A2}=x\left[-\eta c_{A}-\beta \left(c+\theta \lambda h_{1}\right)-\alpha h_{1}+\theta \lambda h_{1}\right]+\left(1-x\right)\left[-\eta c_{A}-\beta \left(\delta c+\theta \lambda h_{2}\right)-\alpha h_{2}+\theta \lambda h_{2}\right]$$

The replication dynamic equation for local government A choosing to fully treat the effluent is as follows:(8)$$F\left(y\right)=\frac{dy}{dt}=y(U_{A1}-{\bar{U}}_{A12})=y\left(1-y\right)\left(U_{A1}-U_{A2}\right)$$

Bringing Equations (6) and (7) into Equation (8), the following can be obtained after calculation:(9)$$F\left(y\right)=y\left(1-y\right)\left\{\theta \left(1-\lambda \right)\left(1-\beta \right)\left[h_{2}\left(1-x\right)+xh_{1}\right]+c_{A}\left(\eta -1\right)\right\}$$

Equation (5) is coupled with Equation (9) to obtain a replicated dynamic system of water polluting enterprises and local government A:(10)$$\left\{ \begin{array}{l} F\left(x\right)=x\left(1-x\right)\left[\theta \left(h_{2}-h_{1}\right)\left(y-y\lambda +\lambda \right)-c\left(1-\delta \right)\right]\\ F\left(y\right)=y\left(1-y\right)\left\{\theta \left(1-\lambda \right)\left(1-\beta \right)\left[h_{2}\left(1-x\right)+xh_{1}\right]+c_{A}\left(\eta -1\right)\right\} \end{array}\right.$$

The joint derivative of Equation (10) is set to zero, and the five local steady state points of the game behavior of water polluting enterprises and local government A are derived separately as follows: O(0,0), A(1,0), B(1,1), C(0,1), $D\left(x^{*},y^{*}\right)$, where $x^{*}=\frac{c_{A}\left(\eta -1\right)+\theta h_{2}\left(1-\lambda \right)\left(1-\beta \right)}{\theta \left(h_{2}-h_{1}\right)\left(1-\lambda \right)\left(1-\beta \right)}\;$,$y^{*}=\frac{c\left(1-\delta \right)-\lambda \theta \left(h_{2}-h_{1}\right)}{\theta \left(h_{2}-h_{1}\right)\left(1-\lambda \right)}$. O(0,0) and B(1,1) are the evolutionary steady state, which correspond to the set of strategies of polluting firms and local government A as follows: {not complete treatment of sewage, not strictly enforcing the RCS} and {complete treatment of sewage, strictly enforcing the RCS}. Figure 1 depicts the dynamic evolution of the game between polluters and local government A.

From Figure 1, we can see that the evolutionary game process between polluters and local government A has four evolutionary paths under two steady state strategy points. On the right side of the ADC, the system evolves to the stable point B(1,1), namely the strategy set {complete treatment of sewage, strictly enforcing the RCS}, whereas on the left side of the ADC, the system evolves to the stability point O(0,0), namely the strategy set {not complete treatment of sewage, not strictly enforcing the RCS}.

It can be seen that the manner in which the game process between polluters and local government A evolves is influenced by two factors: the initial state of the system and the relative position of the local equilibrium point D. When the initial game state of both parties is in the region of ADCO, the stabilization strategy of the game between polluters and local government A evolves to the “Prisoner’s Dilemma”, and both parties do not perform the sewage improvement work completely as the stabilization strategy, and reach the equilibrium state temporarily. If the initial game state of both parties is in the region of ADCB, the stabilization strategy of the game between polluters and local government A evolves to the “Pareto optimal”, and both parties perform the sewage improvement work completely as the stabilization strategy and reach the optimal equilibrium state. The above analysis shows that there are two possible final results of the game between polluters and local government A. Then, the choice of the evolutionary path and direction of the game between the two sides also depends on the relative position of point D. The relative position of point D directly controls the size of the area between region ADCO and region ADCB. When the area of the ADCO interval is larger than the area of the ADCB interval, the system will evolve toward the stable point O(0,0); when the area of the ADCO interval is smaller than the area of the ADCB interval, the system will evolve toward the stable point B(1,1); when the area of the ADCO interval is equal to the area of the ADCB interval, there is uncertainty in the evolution method of the system.

In order to further analyze the factors influencing the evolutionary game path between polluters and local government A, it is necessary to discuss the effect of parameter changes at the local equilibrium point D on the size of the regional ADCB interval, where the size of the ADCO interval is $\left(x^{*}+y^{*}\right)/2$. Table 2 measures the effect of parameter changes at point D on the area of the ADCB interval.

From Table 2, it can be seen that the evolutionary game factors affecting the strategy of enterprise treatment of sewage and local government A implementation of the RCS are the cost of complete treatment of sewage by enterprises, the cost of the RCS implementation by local governments, the strength of treatment of sewage by polluting enterprises, the rate of sewage discharge charges, the weight of economic indicators in the performance appraisal of local governments, the proportion of the increase in the cost of treatment of sewage due to the lack of strict implementation of the RCS by local governments, and the amount of water discharged by enterprises. Among these, the strength of the RCS implementation does not have a fixed direction of influence on the game strategy of the system. In general, as a result of reducing the cost of complete sewage treatment for polluting enterprises, the cost of RCS enforcement by local governments, and the weight of the increase in the cost of sewage treatment due to relaxing the RCS implementation (to some extent, this is also the cost of sewage control for local governments), the change in the value of point D leads to an increase in the area of regional ADCB, and the strategy of the dynamic game between sewage enterprises and local governments A is more likely to evolve in the {complete treatment of sewage, strictly enforcing the RCS} direction of evolution.

For sewage companies, the treatment of sewage is directly related to the cost of action. In the pursuit of cost minimization or profit maximization, any increase in costs needs to be carefully considered. The reduction in the cost of complete treatment of sewage gives companies an opportunity to choose the treatment of pollution, which, in line with the relevant policies, will make companies choose the complete treatment of sewage strategy.

Regarding local governments, under the Chinese decentralized system, the characteristics of political centralization and economic decentralization make local governments pursue the maximization of local economic interests under the incentive of certain political goals. Similarly, the reduced cost of RCS implementation will motivate local governments to choose to strictly implement RCS to accomplish wastewater treatment goals. The reduction in the proportion of economic indicators in the performance appraisal of local governments (correspondingly, the proportion of other non-economic appraisal indicators will increase) will lead local governments to shift the focus of local affairs slightly to non-economic affairs, such as water pollution control, and increase the possibility that local governments will tend to strictly implement RCS.

The stabilization strategy of the dynamic game between the discharging enterprises and the local government A is that the enterprises themselves increase their efforts to treat sewage, while the local government chooses to strictly enforce the RCS strategy. The increase in the sewage charge rate of enterprises increases their sewage costs, and after reaching a certain threshold, enterprises prefer to completely treat their sewage instead of incurring the more costly sewage charge, forcing both sides of the game system to choose the optimal set of stable strategies [14]. At present, in many parts of China, local governments impose fines on sewage discharge enterprises; although the amount varies, basically there is an upper limit. For example, the upper limit of fines before 2020 was RMB 1 million. If the fine for sewage discharge is lower than the cost of water purification, then based on the principle of cost minimization, the enterprise will choose not to implement water purification measures.

For firms, the cost of sewage charges becomes smaller after complete treatment of sewage and they prefer to choose the complete treatment strategy. The stable state of the game between enterprises and local government A is the optimal strategy set. The greater the volume of sewage discharged by enterprises choosing incomplete treatment, the more fees are paid at a constant sewage charge rate, and the dynamic game between polluting enterprises and local government A results in both parties gradually choosing the optimal strategy of {complete treatment of sewage, strictly enforcing the RCS}.

## 3. Evolutionary Game among Local Governments

### 3.1. Model Assumptions and Notation

In order to reasonably set up the evolutionary game model among local governments, all the assumptions in Section 2.1. are also made. In addition to the research hypotheses in the previous section, this study also sets up the following elements based on the actual RCS implementation [4]: ① Based on cost minimization requirements, local governments strictly enforce the RCS with the goal of meeting the water quality standards set by higher levels of government. ② Local governments will be rewarded by higher governments for strict implementation of the RCS, or penalized and subject to losses. ③ It is assumed that information on whether local governments are strictly enforcing the RCS is fully available.

$c_{B}$ is the implementation cost incurred by the local government B’s strict implementation of the RCS, including operating costs and water pollution control costs. Q is a material and spiritualincentive for local governments to strictly implement the RCS. For example, in some areas, the material reward for the assessment of the River Chief System is cash; most of the spiritualrewards relate to the holding of a commendation meeting, or being given the title of “excellent River Chief”. L is the penalty for not strictly implementing the RCS, which results in a loss to the government. G and $G^{\prime }$ are the economic losses of local governments A and B after strict implementation of the RCS, respectively. $w_{A}$ and $w_{A}{}^{\prime }$ are the externality effect coefficients of local government A’s strict and lax implementation, respectively, of the RCS on the water environment of local government B. $w_{B}$ and $w_{B}{}^{\prime }$ are the externality effect coefficients of local government B’s strict and lax implementation, respectively, of the RCS on the water environment of local government A. $H_{A}$ and $H_{A}{}^{\prime }$ are the sewage discharges under local government A’s strict and lax implementation, respectively, of the RCS within the jurisdiction, and $H_{A}{}^{\prime }>H_{A}$. $H_{B}$ and $H_{B}{}^{\prime }$ are the sewage discharges under local government B’s strict and lax implementation, respectively, of the RCS within the jurisdiction, and $H_{B}{}^{\prime }>H_{B}$. $\lambda^{\prime }$ is the strength coefficient of the RCS implementation of local government B and $0<\lambda^{\prime }\leq 1$. $\eta^{\prime }$ is the coefficient of the increased execution cost caused by the greater sewage discharge when the local government B does not strictly implement the RCS, and $\eta^{\prime }>1$. z is the proportion of River Chiefs at all levels within the local government B that choose to strictly implement the RCS.

### 3.2. Model Analysis

The RCS implementation game between local governments is a repetitive game process in which officials are randomly paired, learn from each other, and influence each other. This can be simulated using a replicated dynamic game model. When local government A chooses to strictly implement the RCS, a certain percentage of River Chiefs in neighboring local government B choose to fully implement the RCS for the better water environment. If local government A chooses not to strictly implement RCS, a certain percentage of River Chiefs in local government B may also choose to follow suit.

If local government A and B choose to strictly implement the RCS at the same time, local government A will incur implementation costs $c_{A}$, a small amount of sewage discharges appearing in the loss of water environmental indicators assessment $\alpha H_{A}$, economic performance loss βG, and the local government B to A water quality externality loss $\alpha w_{B}H_{B}$. This loss has the same impact on its environmental indicators assessment, and the benefits of reasonable sewage charges $\theta H_{A}$ and incentives Q. At this point, the local government B payment matrix is set up with the same considerations as the local government A.

When the local government A chooses to strictly enforce the RCS, while local government B chooses not to strictly enforce the RCS, local government A will incur implementation costs, loss of water environmental indicators, loss of economic performance, loss of externalities $w_{B}{}^{\prime }H_{B}{}^{\prime }$, and gain from reasonable sewage charges. At this point, due to the externalities of local government B, local government A will not be rewarded for strict implementation of the RCS. However, the local government B will be punished and incur the loss L.

If both sides do not strictly implement the RCS, each side suffers from the externalities of the other, in addition to the normal cost–benefit payoff, and the penalties imposed on them by the higher government. Repeatedly, the game is played by randomly selecting participants in two groups of local government A and water discharge enterprises in its jurisdiction. In the 2 × 2 asymmetric repeated game, the payoff matrix of its stage game is shown in Table 3.

Suppose the proportion of River Chiefs at all levels in the jurisdiction of local government B who choose to strictly implement the RCS is z. This study uses a replicated dynamic model to simulate a finite rational repeated game process between local governments A and B. The steps used and the calculation method are the same as the game model analysis of local government and enterprises. The five local steady state points of game behavior among local governments are O(0,0), A(1,0), B(1,1), C(0,1), $D\left(x_{D},y_{D}\right)$, where $x_{D}=\frac{c_{A}\left(1-\eta \right)+\left(\theta -\alpha \right)\left(H_{A}{}^{\prime }-H_{A}\right)+\left(1-\lambda \right)\beta G-L}{Q}$, $y_{D}=\frac{c_{B}\left(1-\eta^{\prime }\right)+\left(\theta -\alpha \right)\left(H_{B}{}^{\prime }-H_{B}\right)+\left(1-\lambda^{\prime }\right)\beta G^{\prime }-L}{Q}$. O(0,0) and B(1,1) are the steady state. Figure 2 depicts the evolution of the game among local governments.

As seen in Figure 2, the evolutionary game process between local governments A and B also has four evolutionary paths. The fold ADC is the critical line for the evolution of the different states of the whole system, and the system evolves toward the point in the right region of the fold ADC, the ADCB interval, while the system evolves toward the point in the left region of the fold ADC, the ADCO interval. When local governments A or B do not strictly enforce the RCS, it leads to a sharp increase in implementation cost, and local governments tend to set the strategy of {strictly enforcing the RCS, strictly enforcing the RCS} as a way to reach the equilibrium state.

It can be seen that the game process between local governments is influenced by two factors: the initial state of the system and the relative position of the local equilibrium point D. When the initial state of the game is in the region ADCO, the stabilization strategy of the game between local governments evolves to the “Prisoner’s Dilemma”, and both parties do not strictly implement the RCS as the stabilization strategy, and temporarily reach the equilibrium state. If the initial game state of both parties is in the region ADCB, the stabilization strategy of the local government game evolves to “Pareto optimal”, and both parties strictly implement the RCS as the stabilization strategy to reach the optimal equilibrium state. The management of water quality in the bordering areas is coordinated by the higher-level government, and joint meetings of experts at all levels are held to discuss and formulate strategies.

The above analysis shows that the relative position of point D directly controls the size of the interval area between region ADCO and region ADCB. When the interval area of ADCO is larger than the interval area of ADCB, the system will evolve toward the point D; when the interval area of ADCO is smaller than the interval area of ADCB, the system will evolve toward the point D; when the interval area of ADCO is equal to the interval area of ADCB, the evolution of the system direction is unknown. Table 4 measures the effect of the parameter change at point D on the area of the ADCB interval.

As seen in Table 4, either reducing the cost of RCS implementation for local government A or local government B can induce a change in the relative position of the D-point values to evolve in the direction of {strictly enforcing the RCS, strictly enforcing the RCS}. On the one hand, local governments not strictly enforcing the RCS will reduce some of the costs of regulation; however, on the other hand, local governments seeking to maximize financial benefits will choose to strictly enforce RCS because the increase in the cost of treating sewage is much greater than the economic loss of treating pollution. Of course, under the premise that other influencing factors remain unchanged, if either local government A or B strictly enforce the RCS, enterprises will have to treat sewage. Thus, the smaller the loss to the local economy, the more likely the local government will choose to strictly enforce the RCS.

Under the strict RCS assessment, local governments will choose to exchange a smaller economic loss for a better water quality environment and to improve social welfare. Increasing the enforcement of the RCS by local governments means spending more human, material, financial and other resources, leading to an increase in the area of ADCB, and the choice of the dynamic game strategy set among local governments gradually evolves in the direction of {strictly enforcing the RCS, strictly enforcing the RCS}. In China’s environmental governance, the role of public participation is not reflected. Even if there are channels for participation, there are relatively few people who actually implement it. This is also a point that needs to be paid attention to in China’s environmental reform in the future.

The increase in the proportion of water quality and environmental indicators in the performance appraisal of local governments A and B by the higher-level government, or the decrease in the proportion of economic indicators in the appraisal, are conducive to the evolution of the strategy set of the dynamic game between local governments in the direction of {strictly enforcing the RCS, strictly enforcing the RCS}. If local governments continue to ignore the local water quality, their overall political appraisal will be affected and the political demands of local governments will be hindered by the failure to meet the water quality index appraisal.

The rewards and punishments applied by higher-level governments to lower-level governments for RCS implementation also affect the choice of game strategies. Both the increase in the material and moral rewards, and the punishment, will lead to the evolution of the game strategy set among local governments toward the “Pareto optimal” direction. Under the Chinese decentralized system, the political promotion of local officials depends on the performance evaluation of the higher-level government. At this stage, according to the relevant documents of local governments in various provinces and cities, most of them have set up a “one-vote veto” system for RCS appraisal arrangements, and the deterioration in the water quality due to improper implementation of the RCS means the cancellation of the promotion opportunities of the local personnel, or their demotion; worse, they will be held accountable for life. This assessment mechanism is designed to eliminate the possibility that some local officials are only concerned about their personal political future and economic interests, and thereby approve projects with significant water pollution hazards. In fact, the positions of China’s River Chiefs are concurrently held by the main leaders of each region, and they all make their own decisions when setting goals according to the specific conditions of their respective regions. Therefore, the impact of individual abilities will be relatively small.

From Table 4, it can be seen that the reduction in effluent discharge after local governments do not strictly implement the RCS increases the ADCB area. This means that the local government will tend to choose the strategy of strict implementation of the RCS if the economic gain of the local government due to sewage charges is reduced, or less sewage discharge makes it easier for the local government to accomplish the goal of the RCS, provided that other factors remain unchanged. The analysis of the results in Table 4 and the setup of the dynamic game model among local governments shows that the externalities of sewage discharge do not affect the evolutionary direction and path of the whole game system, regardless of whether the local government strictly implements the RCS.

## 4. Conclusions

The full implementation of RCS is an inevitable requirement for promoting ecological sustainability and an effective measure to solve China’s complex water problems. The choice of the RCS implementation strategy by local governments directly affects the quality of the water environment in the watershed. Therefore, this study analyzes the evolutionary game process between wastewater governance of enterprises and RCS implementation by local governments, and between local governments, based on the reform context of the RCS.

It is found that reducing the cost of complete sewage treatment by enterprises, the cost of RCS implementation by local governments, and the amount of sewage discharged by enterprises after complete sewage treatment will lead to the evolution of the strategy set in the direction of {complete treatment of sewage, strictly enforcing the RCS}. Increasing the enterprises’ own efforts to treat wastewater, and the rate of charging enterprises for wastewater discharge will also lead to the evolution of the strategy set in the direction of {complete treatment of sewage, strictly enforcing the RCS}.

The game analysis of RCS implementation among local governments found that reducing the cost of RCS implementation and the proportion of economic indicators in the assessment of local governments, and increasing the implementation intensity of the RCS, the proportion of water quality and environmental indicators in the performance assessment, and the assessment strength will lead to the evolution of the strategy set in the direction of {strictly enforcing the RCS, strictly enforcing the RCS}. The external effects of sewage discharge will not have an impact on the evolution of the whole game system.

## Figures and Tables

**Figure 1 ijerph-19-01961-f001:**
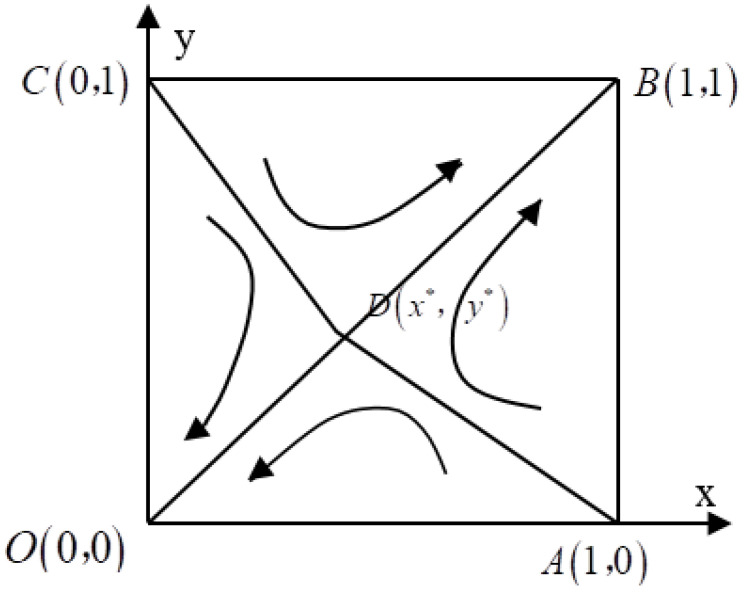
Phase diagram of the evolutionary game between polluters and local government A.

**Figure 2 ijerph-19-01961-f002:**
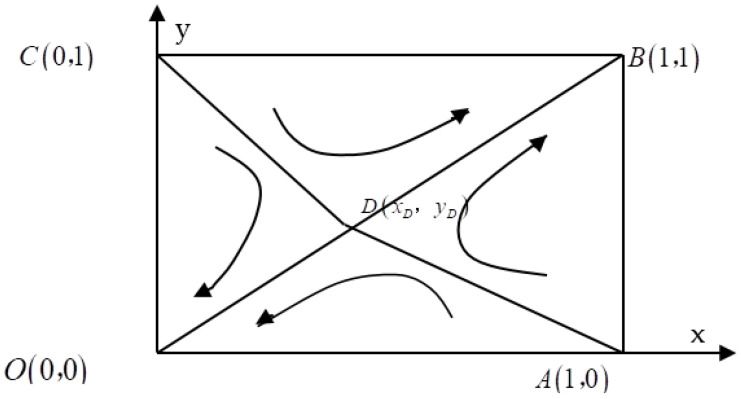
Phase diagram of the evolutionary game among local governments.

**Table 1 ijerph-19-01961-t001:** Game payment matrix between local government and water polluting enterprises.

	Local Government A Strict Implementation of RCS	Local Government A Not Strict Implementation of RCS
Enterprises completely treat sewage	$-c-\theta h_{1},-c_{A}-\beta \left(c+\theta h_{1}\right)-\alpha h_{1}+\theta h_{1}$	$-c-\theta \lambda h_{1},-\eta c_{A}-\beta \left(c+\theta \lambda h_{1}\right)-\alpha h_{1}+\theta \lambda h_{1}$
Enterprises do not completely treat sewage	$-\delta c-\theta h_{2},-c_{A}-\beta \left(\delta c+\theta h_{2}\right)-\alpha h_{2}+\theta h_{2}$	$-\delta c-\theta \lambda h_{2},-\eta c_{A}-\beta \left(\delta c+\theta \lambda h_{2}\right)-\alpha h_{2}+\theta \lambda h_{2}$

**Table 2 ijerph-19-01961-t002:** Effect of parameter changes on the evolutionary strategy between polluters and local government.

Parameter Change	Change in Point D	ADCB Interval Area Change	Game Strategy
c↓	y*↓	↑	{complete treatment of sewage, strictly enforcing the RCS}
c_A_↓	x*↓	↑	{complete treatment of sewage, strictly enforcing the RCS}
δ↑	y*↓	↑	{complete treatment of sewage, strictly enforcing the RCS}
λ↑	x*↑; y*↓	indefinite	indefinite
θ↑	x*↓; y*↓	↑	{complete treatment of sewage, strictly enforcing the RCS}
h_2_↑	x*↓; y*↓	↑	{complete treatment of sewage, strictly enforcing the RCS}
h_1_↓	x*↓; y*↓	↑	{complete treatment of sewage, strictly enforcing the RCS}
β↓	x*↓	↑	{complete treatment of sewage, strictly enforcing the RCS}
η↓	x*↓	↑	{complete treatment of sewage, strictly enforcing the RCS}

**Table 3 ijerph-19-01961-t003:** Game payment matrix between the local governments.

	B Strictly Implement the RCS	B Not Strictly Implement the RCS
A strictly implements the RCS	$\begin{array}{l} -c_{A}+\theta H_{A}-\alpha H_{A}-\beta G-\alpha w_{B}H_{B}+Q,\\ -c_{B}+\theta H_{B}-\alpha H_{B}-\beta G^{\prime }-\alpha w_{A}H_{A}+Q \end{array}$	$\begin{array}{l} -c_{A}+\theta H_{A}-\alpha H_{A}-\beta G-\alpha w_{B}{}^{\prime }H_{B}{}^{\prime },\\ -\eta^{\prime }c_{B}+\theta H_{B}{}^{\prime }-\alpha H_{B}{}^{\prime }-\beta \lambda^{\prime }G^{\prime }-\alpha w_{A}H_{A}-L \end{array}$
A does not strictly implement the RCS	$\begin{array}{l} -\eta c_{A}+\theta H_{A}{}^{\prime }-\alpha H_{A}{}^{\prime }-\beta \lambda G-\alpha w_{B}H_{B}-L,\\ -c_{B}+\theta H_{B}-\alpha H_{B}-\beta G^{\prime }-\alpha w_{A}{}^{\prime }H_{A}{}^{\prime } \end{array}$	$\begin{array}{l} -\eta c_{A}+\theta H_{A}{}^{\prime }-\alpha H_{A}{}^{\prime }-\beta \lambda G-\alpha w_{B}{}^{\prime }H_{B}{}^{\prime }-L,\\ -\eta^{\prime }c_{B}+\theta H_{B}{}^{\prime }-\alpha H_{B}{}^{\prime }-\beta \lambda^{\prime }G^{\prime }-\alpha w_{A}{}^{\prime }H_{A}{}^{\prime }-L \end{array}$

**Table 4 ijerph-19-01961-t004:** Effect of parameter changes on the evolutionary strategy among local governments.

Parameter Change	Change in Point D	ADCB Interval Area Change	Game Strategy
c_A_↓	z*↓	↑	{complete treatment of sewage, strictly enforcing the RCS}
c_B_↓	y*↓	↑	{complete treatment of sewage, strictly enforcing the RCS}
λ↑	z*↓	↑	{complete treatment of sewage, strictly enforcing the RCS}
λ′↑	y*↓	↑	{complete treatment of sewage, strictly enforcing the RCS}
θ↓	y*↓; z*↓	↑	{complete treatment of sewage, strictly enforcing the RCS}
α↑	y*↓; z*↓	↑	{complete treatment of sewage, strictly enforcing the RCS}
β↓	y*↓; z*↓	↑	{complete treatment of sewage, strictly enforcing the RCS}
η↑	z*↓	↑	{complete treatment of sewage, strictly enforcing the RCS}
η′↑	y*↓	↑	{complete treatment of sewage, strictly enforcing the RCS}
H_A_′↓	z*↓	↑	{complete treatment of sewage, strictly enforcing the RCS}
H_A_↑	z*↓	↑	{complete treatment of sewage, strictly enforcing the RCS}
H_B_′↓	y*↓	↑	{complete treatment of sewage, strictly enforcing the RCS}
H_B_↑	y*↓	↑	{complete treatment of sewage, strictly enforcing the RCS}
Q↑	y*↓; z*↓	↑	{complete treatment of sewage, strictly enforcing the RCS}
L↑	y*↓; z*↓	↑	{complete treatment of sewage, strictly enforcing the RCS}
G↓	z*↓	↑	{complete treatment of sewage, strictly enforcing the RCS}
G′↓	y*↓	↑	{complete treatment of sewage, strictly enforcing the RCS}

## Data Availability

Not applicable.

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
