# Peer review of "Game Analysis of the Evolution of Local Government’s River Chief System Implementation Strategy"

_ijerph, 2022, doi:10.3390/ijerph19041961_

Round 1
Reviewer 1 Report
This paper focuses on the choice implementation strategies used for local government related activities to evaluate the quality of regional waster environement. The gane model evaluation has been conducted and results has presented well. The study is interesting. However, there are some suggestions for imporving the quality of the paper as follows:
(1) What are the significance of the study? The athoers might consider to elaborate this model analysis and how does it contribute to the theory and practice.
(2) What are the practical implications of this study?
(3) What are the assumption made for model development using this case application? What are the limitations of the model?
(4) What are the future studies?
Author Response
Dear editor and expert,
Thank you for your useful comments and suggestions on the content of our manuscript. We have modified the manuscript accordingly, and the detailed corrections are listed below point by point:
Reviewer #1:
1) What are the significance of the study? The athoers might consider to elaborate this model analysis and how does it contribute to the theory and practice.
√ Existing literature rarely analyzes the impact of strategies between enterprises and local governments, and between local governments from the perspective of RCS. In order to effectively model these dynamic and long-term game relationships among the implementation of RCS, the article uses an evolutionary game model to explore the evolutionary process of RCS implementation decisions between enterprises and local governments, and between local governments. The article complements the existing literature from a game theory perspective.
2) What are the practical implications of this study?
√ The exploration for these strategic behaviors is a necessary study in the process of RCS reform, which helps to reveal the nature of water pollution management in China from one side and improve the efficiency of the implementation of the RCS.
3) What are the assumption made for model development using this case application? What are the limitations of the model?
√ We explain in detail in the assumptions section: In order to reasonably set up the evolutionary game model between local governments and water discharge enterprises, drawing on existing studies[12],[13], the following assumptions are set up in this study. (I)The RCS assessment system is the same for areas under the jurisdiction of the same provincial government. (II)Under the strict arrangement of RCS, local governments are required to treat and improve existing waters and other various tasks in addition to supervising water discharge enterprises in their jurisdictions. There is no possibility for local governments not to implement the RCS work arrangement, or they will face huge political and economic penalties. Likewise, there is no possibility for water discharge enterprises in the jurisdiction to not treat their own sewage at all. (III)The goal of water discharge enterprises based on the economic goal of cost minimization and complete treatment of sewage is to meet the water pollution discharge standards set by the government. (IV)All parameters are in the same policy cycle during the study sample period and will not change over time. (V)The strict implementation of the RCS is able to make the water pollution in the jurisdiction controlled and the water quality significantly improved. (VI)It is assumed that the regulatory costs incurred by local governments when they do not fully implement the RCS (including enforcement costs, costs of treating effluent, etc.) are greater than the costs associated with completing strict implementation of the RCS (at this point, the costs incurred by local governments for deregulation resulting in more effluent treatment are greater than the enforcement costs reduced by less strict regulation).
4) What are the future studies?
√ In the future, on the basis of in-depth enterprise research, we will continue to study how the reform of the river chief system affects the behavior of water polluting enterprises. Of course, the framework also continues to be based on game theory, with detailed discussions on the details of corporate behavior.
The manuscript has been resubmitted to your journal. We look forward to your positive response.
Best wishes,
Ruide Tu
Reviewer 2 Report
Review of Manuscript
Overall, it is an interesting and innovative approach (application of game theory), for which I congratulate the authors.
I would like to call attention to some aspects around the implementation of the RCS that would, in my opinion, be relevant in a research work, and an academic publication:
Lines 51-52 – “Existing literature rarely analyses the impact of strategies between enterprises and local governments, and between local governments from the perspective of RC” – this is an innovative and useful contribution of this paper.
Line 58 – “The characteristics of the institutional setting and task arrangement determine that the RCS is more authoritative and binding than previous water pollution management policies in China, and the policy is more sustainable” – This statement would require a clearer context justification and explanation – is it because of cultural, institutional, or other reasons?
Line 84 – “There is no possibility for local governments not to implement the RCS work arrangement, or they will face huge political and economic penalties” – It would be relevant to understand what political penalties are these? Is the low performance penalized in terms of mandates for example? Will the governmental bodies be replaced? And besides, are there examples of good performance outside this RC system?
Lines 273-276 – “The increase in the sewage charge rate of enterprises increases their sewage costs, and after reaching a certain threshold, enterprises prefer to completely treat their sewage instead of the more costly sewage charge, forcing both sides of the game system to choose the optimal set of stable strategies” - It should be clarified further what are the financial penalties (in % of initial treatment costs) that make the stabilizing point in the model become so attractive to the companies. And, regarding the local governments, what are the economic penalties to their performance.
On the other hand, the threshold (paying or not treatment / paying extra fees) also depends on the kind of pollutants, and treatment requirements… how is this applied?
Line 285 – “in addition to the research hypotheses in the previous section” - What assumption do you refer to? It should be repeated to make it clear for the following argumentation.
Lines 292-293 – “Q is a material and spiritual incentive for local governments to strictly implement RCS” - Whatever interesting is the ‘spiritual dimension’ it should be clearly explained what is it specifically in this context, and how is this incentive effective.
Lines 308 to 313, and beyond - when you present the evolutionary game process between local governments, you refer to the proximity influence factor. It would be also important to understand what mechanism of communication, and collaboration among different Local Governments are in place to support the RCS.
After Lines 366 - while discussing the results, it would be important to understand if there are other factors that influence the local governments’ decisions, besides the economic incentive, e.g., the local community’s involvement, and needs, health issues regarding specific pollutants, better water quality for some uses, etc.
Even when you refer: “Under the strict RCS assessment, local governments will choose to exchange a smaller economic loss for a better-quality water quality environment and improve social welfare”, it is not very clear what role the citizens, and communities’ involvement play in these decisions. This may be relevant if you want to discuss the RCS in other contexts outside China.
In line 397 - there is a further clarification of how the ‘penalties’ can be applied - there is no context justifications in place for cases where the responsibility can be also of other external factors that overcome the individual capacity of resolution.
In line 410 - I would strongly suggest contextualizing, as something that is not ‘inevitable’, and effective measure, outside the China context. Likewise, the final conclusion in line 426 deserves also to be framed and contextualized within the very specific case of China.
Good luck with the review process, and publication.
Best wishes form the Reviewer
Author Response
Dear editor and expert,
Thank you for your useful comments and suggestions on the content of our manuscript. We have modified the manuscript accordingly, and the detailed corrections are listed below point by point:
Reviewer #2:
1) Line 58 – “The characteristics of the institutional setting and task arrangement determine that the RCS is more authoritative and binding than previous water pollution management policies in China, and the policy is more sustainable” – This statement would require a clearer context justification and explanation – is it because of cultural, institutional, or other reasons?
√ We add some content in the paper “most of the performance assessments of River Chiefs in each region are linked to the effectiveness of the implementation of the RCS, which making them much more politically risky to advance in the position if they focus only on economic growth and ignore the water pollution, so the local officials are more motivated to supervise sewage discharging enterprises in their jurisdictions based on political promote. In addition to the effect of the implementation of RCS is linked to the evaluation of officials' promotion, most regions also impose economic or political punishments, such as demotion and legal sanctions, for officials with poor implementation of the river chief system. This gives the policy system guarantee of RCS.”
2) Line 84 – “There is no possibility for local governments not to implement the RCS work arrangement, or they will face huge political and economic penalties” – It would be relevant to understand what political penalties are these? Is the low performance penalized in terms of mandates for example? Will the governmental bodies be replaced? And besides, are there examples of good performance outside this RC system?
√ We have added some explanations where appropriate “… Of course, if the economic benefits obtained by the government from not implementing RCS are greater than implementing reforms, and government officials do not consider penalties such as promotion or demotion, they may also choose not to implement RCS.”
3) Lines 273-276 – “The increase in the sewage charge rate of enterprises increases their sewage costs, and after reaching a certain threshold, enterprises prefer to completely treat their sewage instead of the more costly sewage charge, forcing both sides of the game system to choose the optimal set of stable strategies” - It should be clarified further what are the financial penalties (in % of initial treatment costs) that make the stabilizing point in the model become so attractive to the companies. And, regarding the local governments, what are the economic penalties to their performance.
√ We have add the explanation for it “At present, in many parts of China, local governments impose fines on sewage discharge enterprises, and the amount varies, but basically there is an upper limit. For example, the upper limit of fines before 2020 is 1 million RMB. If the fine for sewage discharge is lower than the cost of water purification, then based on the principle of cost minimization, the enterprise will choose not to implement water purification measures…..”
4) On the other hand, the threshold (paying or not treatment / paying extra fees) also depends on the kind of pollutants, and treatment requirements… how is this applied?
√ Before 2020, the upper limit of fines imposed by the environmental protection department on illegal polluting enterprises is 1 million yuan, and a single polluting behavior can only be punished once. In the past few decades, only a handful of companies have been fined more than 1 million yuan by the environmental protection department. In many places, the following scenario has even occurred: At the beginning of the year, polluting companies took money to the environmental protection department, saying that this was a one-year fine, and once the payment was enough, the environmental protection department would not use it for law enforcement throughout the year.
5) Line 285 – “in addition to the research hypotheses in the previous section” - What assumption do you refer to? It should be repeated to make it clear for the following argumentation.
√ We have made a more precise description in the article “(I)The RCS assessment system is the same for areas under the jurisdiction of the same provincial government. (II)Under the strict arrangement of RCS, local governments are required to treat and improve existing waters and other various tasks in addition to supervising water discharge enterprises in their jurisdictions. There is no possibility for local governments not to implement the RCS work arrangement, or they will face huge political and economic penalties. Likewise, there is no possibility for water …... ”.
6) Lines 292-293 – “Q is a material and spiritual incentive for local governments to strictly implement RCS” - Whatever interesting is the ‘spiritual dimension’ it should be clearly explained what is it specifically in this context, and how is this incentive effective.
√ We revised the texts carefully and add some content in the article “For example, in some areas, the material reward for the assessment of the river chief system is cash; most of the spiritual rewards are to hold a commendation meeting, or to give the title of "excellent river chief".”
7) Lines 308 to 313, and beyond - when you present the evolutionary game process between local governments, you refer to the proximity influence factor. It would be also important to understand what mechanism of communication, and collaboration among different Local Governments are in place to support the RCS.
√ We revised the texts carefully in the article, such as “It can be seen that the game process between local governments is influenced by two factors: the initial state of the system and the relative position of the local equilibrium point D. When the initial state of the game is in the region ADCO, the stabilization strategy of the game between local governments evolves to the "Prisoner's Dilemma", and both parties do not strictly implement the RCS as the stabilization strategy, and temporarily reach the equilibrium state. If the initial game state of both parties is in the regional ADCB, the stabilization strategy of the local government game evolves to "Pareto optimal", and both parties strictly implement the RCS as the stabilization strategy to reach the optimal equilibrium state. The management of water quality in the bordering areas is coordinated by the higher-level government, and the joint meetings of experts at all levels are held to discuss and formulate strategies.……”.
8) After Lines 366 - while discussing the results, it would be important to understand if there are other factors that influence the local governments’ decisions, besides the economic incentive, e.g., the local community’s involvement, and needs, health issues regarding specific pollutants, better water quality for some uses, etc.
√ Given the limitations of evolutionary game models, the paper does not consider these factors and add them to the model. However, these factors raised by external audit experts are good points for discussion, and we will analyze them in detail in the next article devoted to the reform of the river chief system and corporate behavior. Thanks again for your advice and help.
9) Even when you refer: “Under the strict RCS assessment, local governments will choose to exchange a smaller economic loss for a better-quality water quality environment and improve social welfare”, it is not very clear what role the citizens, and communities’ involvement play in these decisions. This may be relevant if you want to discuss the RCS in other contexts outside China.
√ Yes, in China's environmental governance, the role of public participation has not been reflected. Even if there are channels for participation, there are relatively few people who actually implement it. This is also a point that needs to be paid attention to in China's environmental reform in the future.
10) In line 397 - there is a further clarification of how the ‘penalties’ can be applied - there is no context justifications in place for cases where the responsibility can be also of other external factors that overcome the individual capacity of resolution.
√ In fact, China's River Chiefs are concurrently held by the main leaders of each region, and they all make their own decisions when setting goals according to the specific conditions of their respective regions. Therefore, the impact of individual abilities will be relatively small. We have revised this in the zrticle.
11) In line 410 - I would strongly suggest contextualizing, as something that is not ‘inevitable’, and effective measure, outside the China context. Likewise, the final conclusion in line 426 deserves also to be framed and contextualized within the very specific case of China.
√ In fact, the reform of the river chief system in China has just begun, and the reform measures in each region have their own characteristics. In China's political background, the river chief system can be implemented relatively effectively. After all, China implements a system of political centralization and economic decentralization. Therefore, apart from the institutional background of China, different conclusions may emerge.
The manuscript has been resubmitted to your journal. We look forward to your positive response.
Best wishes,
Ruide Tu
